# The Synergistic Effects of Ultrafine Slag Powder and Limestone on the Rheology Behavior, Microstructure, and Fractal Features of Ultra-High Performance Concrete (UHPC)

**DOI:** 10.3390/ma16062281

**Published:** 2023-03-12

**Authors:** Congqi Luan, Qingchun Yang, Xinru Lin, Xin Gao, Heng Cheng, Yongbo Huang, Peng Du, Zonghui Zhou, Jinbang Wang

**Affiliations:** 1Shandong Provincial Key Laboratory of Preparation and Measurement of Building Materials, University of Jinan, Jinan 250022, China; luancq0822@163.com (C.L.);; 2School of Materials Science and Engineering, University of Jinan, Jinan 250022, China; 3School of Civil and Architectural Engineering, University of Jinan, Jinan 250022, China

**Keywords:** UHPC, synergistic effect, fractals, ultrafine slag powder, limestone

## Abstract

This study investigated the effect of the interaction between ultrafine slag powder (USL) and limestone (LS) on the rheology behavior, microstructure, and fractal features of UHPC. The results indicated that B2 with mass ratio of 2:1 between the USL and LS obtained the highest compressive strength and the lowest yield stress. The combination of the USL and LS facilitated the cement hydration, ettringite, and monocarboaluminate (Mc) formation, as well as the increase in the polymerization of the C–S–H. The synergistic action between the USL and LS refined the pore structure due to the formation of the Mc, compensating for the consumption of the CH by the pozzolanic reaction, which provided a denser microstructure in the UHPC. The fractal dimension (Ds) of the UHPC was strongly related to the concrete pore structures and the compressive strength, which demonstrated that a new metric called the Ds value may be used to assess the synergistic effect of the UHPC.

## 1. Introduction

UHPC is regarded as the most promising cementitious material in the concrete industry [1]. However, the large amount of cementitious material required for the preparation of UHPC has increased global carbon dioxide emissions [2,3]. There is a high potential to reduce the use of a cement clinker and CO_2_ emissions by adding supplementary cementitious materials (SCMs). Furthermore, incorporating a proper proportion of SCMs in the cement can improve the workability and increased the strength and the durability of the concrete [4,5].

Limestone (LS) powder with an environmental impact has been used in cement-based materials for a long time [2]. The effects of LS on the performance of the concrete, including the workability [6], strength [7,8], dimensional stability [9], and durability [10,11], have been investigated. It is well known that LS promoted the precipitation of the C–S–H and accelerated the clinker hydration [4]. However, LS also reacted with the alumina of the SCMs and showed a positive synergistic effect [12]. The calcite offered from LS reacted with the alumina to form a hemi-carboaluminate (Hc) and monocarboaluminate (Mc) phase, which increased the total volume of the hydration products and blocked the tiny pores, thus refining the pore structures and enhancing the compressive strength. Suhua Ma et al. [2] and Antoni et al. [13] indicated that incorporating metakaolin (MK) and LS significantly refined the pore structure and increased the compressive strength due to the stimulation of the C_3_S hydration and the ettringite stabilization. Adu-Amankwah et al. [14] demonstrated that there were synergistic effects between LS and the slag. The nucleation effects of LS enhanced the cement hydration and increased the aluminum concentration in the pore solution, which improved the slag hydration. However, Axel Schöler et al. [15] reported the actual reaction content of Al_2_O_3_ in the slag with LS was very low due to the lower activity of the slag, which limited the formation of the Hc and Mc in the concrete.

In contrast to the normal ground granulated blast furnace slag, ultrafine slag powder (USL) had an increased surface area, activity index, and an amorphous form of particles [16], which increased both the hydration rates and the number of pozzolanic reactions and improved its reaction chances with LS [17]. Herein, it is interesting to explore the synergistic action of the USL and LS on the UHPC. Furthermore, the synergistic action between the USL and LS on the rheological behaviors of the UHPC has few reports. Moreover, the USL possessed a higher pozzolanic reaction and a finer particle distribution, which not only affected the rheological properties and influenced the formation and change of the pore structure of the UHPC, but also increased the probability of the reaction with the limestone powder. The USL and LS compounded the ternary cement-based systems and posed a better potential for influencing the pore structure of the UHPC in various ways, including the filling effect, pozzolanic reactions, nucleation effect, and so on, which resulted in a pore structure or pore size distribution of the concrete that was highly complex with irregular features. It was incomplete to evaluate the effect of the synergistic action of the USL and LS on the pore structure of the UHPC using the porosity. To deeply understand the synergistic action on the microstructures of the UHPC, fractal geometry provided an effective way to investigate the complexity and heterogeneity of the microstructures of the UHPC, which also linked the macroscopic performance of the UHPC to their microscopic characteristics [18,19,20]. The complex and irregular pore structure was quantified by the fractal dimension (*D*s) values [21,22,23]. The Ds has been proven to be closely associated with the compressive strength, which was revealed by Jin et al. [24] and Wang et al. [20]. The most direct manifestation of the USL and LS synergistic action in the UHPC is to obtain a denser microstructure and a high compressive strength of the hardened concrete.

However, using the fractal theory to study the synergistic action of the USL and LS on the performance of the UHPC is limited. Hence, the target of this study is to explore the effect on the microstructure and the properties of the UHPC containing the USL and LS and to better understand this synergy effect. The rheological properties and workability of the UHPC incorporating the various ratios of the USL and LS were first investigated. After that, the phase development of the paste was assessed using the X-ray diffraction (XRD) analysis. Additionally, the microstructure morphology and evolution were obtained using SEM and MIP, respectively. Furthermore, the final target is to use the fractal theory to better understand the synergistic action of the USL and LS on the performance of the UHPC.

## 2. Experimental

### 2.1. Materials

The ordinary Portland cement (P·O 52.5, from Shanshui Cement Co., Ltd., Shandong Province, China), ultrafine slag powder (USL, from the building material market and after grinding), and limestone (LS, from the Gongyi Yuan Heng water purification material factory in Henan Province, China) were used in this study. Their compositions are listed in Table 1. The polycarboxylate superplasticizer (SP) and straight steel fibers were used in this study. The particle size distribution, XRD patterns, and SEM for all raw materials were presented in Figure 1, Figure 2 and Figure 3, respectively. The median particle sizes of P·O 52.5, USL, and LS were 10.89 μm, 4.307 μm, and 13.50 μm, respectively. The USL had a finest distribution than the LS and P·O 52.5, and the LS obtained the coarsest distribution. The amorphous phase was observed in the USL and visible peaks of calcium carbonate were observed in LS. Figure 3 shows the SEM pictures of the USL and LS particles. As can be seen in Figure 3, the limestone was mostly cubic or elongated, with sharp angles. The superfine slag powder had smaller, flaky, or plate shaped particles.

### 2.2. Sample Preparation

In this study, the water-to-binder and sand-to-cement ratios were fixed at 0.18 and 1, respectively. Mixtures with 2% steel fibers and different contents of the USL and LS were introduced to the UHPC mixtures, as summarized in Table 2. A superplasticizer (SP) was used to adjust the workability of the UHPC mixture, and its dosage was fixed at 1.5% by a mass of binders. The binder materials were mixed for 3 min, and thus the mixing solution (including water and the superplasticizer) was added. Continue to mix at low speed until the mixture became a paste, followed by rapid mixing for 3 min. Finally, the sand and fiber were added gradually at a low speed and then stirred at high speed for 3 min. Then the mixtures were cast into a 40 mm × 40 mm × 160 mm mold and cured in a standard room (Relative humidity > 90%, 20 °C). After demolding, the UHPC was cured in a standard room for testing at 7 d and 28 d. The paste without the steel fiber and sand was prepared to analyze the hydration reaction and microstructure of the UHPC.

### 2.3. Test Method

The behavior of the rheological was measured by the cement paste. The Malvern rheometer was equipped with an automatic temperature control device and the shear rate control protocol is shown in Figure 4. To eliminate the thixotropy, the slurry was pre-sheared for 20 s and stabilized for 20 s, before undergoing the falling stages, which lasted for 200 s. The experimental temperature remained constant at 20 °C [25]. After it was vibrated 25 times, the mean diameter of the mortar was considered as the flowability, according to standard Chinese GB/T2419-2005. The compressive strength result of the mortar was obtained from six specimens with the loading rate of 2.4 kN/s. The fragment of the paste was ground into powder for XRD and ^29^Si NMR spectroscopy on the designated test day. After passing through a 200-mesh sieve, the powder was heated and dried to a constant weight at 60 °C under a vacuum drying oven.

The powder was pressed into thin sheets for the XRD test, which was performed at 0.2 s per step from 5° to 60° with the CuKα radiation at 40 kV and 40 mA. The SEM observed microstructure of the paste contained the USL or LS. MIP evaluated the pore structure of the paste. The solid-state ^29^Si NMR spectra were obtained from the AVANCE III HD 600 MHz NMR spectrometer at 9.4 T and 79.4 MHz in the magnetic field intensity and magnetic field frequency, respectively.

The fractal features of the concrete pore structure was evaluated using Zhang’s model [20,26]. There is a logarithmic law between the cumulative intrusion work (*Wn*) and the total amount of pores (*V_n_*) injected with mercury [27]. The calculation formula was
(1)$$ln(\frac{W_{n}}{\gamma_{n}^{2}})=D_{s}ln\frac{{V_{n}}^{\frac{1}{3}}}{r_{n}}+lnC$$ where *r_n_* is the smallest pore radius, m; *C* is a regression constant; and *D_s_* is the pore surface fractal dimension.

*Wn* is calculated using Formula (2).
(2)$$w_{n}=\sum_{i=1}^{n}p_{i}\Delta v_{i}$$
where $p_{i}$ represents the pressure at the ⅈth injection of mercury, Pa; $v_{i}$ means the volume of the ⅈth injection of mercury, m^3^; and *n* stands for the amount of mercury injected.

## 3. Result and Discussion

### 3.1. Rheological Properties

The rheological behavior of the UHPC pastes with the different USL or LS content is shown in Figure 5A. For the pure cement paste (Ref), the shear stress increased rapidly with the increasing shear rate, which was typical for swelling fluids. Compared to the Ref, the fluid types of the rheology curves containing the USL or LS remained consistent, which illustrated that the non-linear Herschel–Bulkley (3) model was suitable to fit the non-linear flow conjunction between the shear stress and shear rate, and the parameters are shown in Table 3.
τ = τ_0_ + *k*γ^n^(3)
where τ_0_ represents the yield stress, *k* represents the correction parameter, and *n* represents the paste’s fluid flow index. The fluid flow index n-value of all the mixtures was higher than 1.0, demonstrating that the pastes’ rheological flow curve obtained a shear-thickening response [28,29]. It was further proven that the slurry presented a shear-thickening response, and the apparent viscosity changes are shown in Figure 5B. The viscosity increased continuously with the shear rate, which was consistent with the response of *n* > 1 [30,31]. From Figure 5 and Table 3, the n-values varied between 3.667 and 4.975 in this study. The lowest n-value of 3.667 and the yield stress of 20.822 Pa were obtained from B2 with 20% USL and 10% LS. Furthermore, the USL30 and B6 received the highest n-value of 4.975 and the highest yield stress of 47.204 Pa. At 30% total cement substitution, the yield stress and n-values of the pastes were high, whether LS or the USL replaced the cement. Still, the appropriate ratio of LS to the USL in the mixtures significantly reduced the yield stress and n-values. Since the average particle size of the USL was smaller than that of the cement particles and the average particle size of LS was more extensive than that of the cement particles, their addition improved the distribution of the total cement-based particles. Incorporating 20% USL and 10% LS in the cement, the USL filled the particle space, released the trapped water, and incremented the free water content. The large LS particle extended the skeleton space of the paste, the cement particles filled the particle space of the skeleton space of the LS, and the USL particles filled the space between the cement particles, leading to a decrease in the shear thickening and the yield stress [32,33]. Increasing the USL content increased the adsorption of the water reducing agent and its agglomeration effect, which maintained the positive impact imposed by the increase in free water and influenced the degree of shear thickening [34]. Moreover, when 30% of the cement was replaced by LS, the LS particles increased the skeleton space of the paste, which needed more cement paste to fill and resulted in smaller spaces not being filled [35]. At this time, the disordered arrangement of the particles increased the yield stress of the paste. Consequently, the plastic viscosity could not be given due to the non-linear relationship between the shear stresses and the shear rates. However, the plastic viscosity can been calculated using the De Larrard Equation (4) [36,37]:(4)$$u^{\prime }=\frac{3k}{n+2}\gamma_{max}^{n-1}$$
where $u^{\prime }$ presets the plastic viscosity and $\gamma_{max}$ is the maximum shear strain rate achieved in the test. According to Table 3, B2 obtained the lowest plastic viscosity, which was consistent with the above analysis. The reasonable particle gradation reduced the yield stress and plastic viscosity.

### 3.2. Flowability

Figure 6A illustrates that the slump flowability of the UHPC containing 20% USL and 10% LS (B2) showed a higher fluidity than USL30, B6, and the Ref. B2 obtained the maximum slump flowability of 245 mm, which increased by 48.8%, 32.4%, and 44.1% compared to the Ref, USL30, and B6, respectively. The USL filled the particle space, released the trapped water, and incremented the free water content. The large LS particle extended the skeleton space of the paste, leading to the decrease in the yield stress and the frictional resistance, which improved the mixture flowability [38,39]. The appropriate ratio of LS to the USL in the mixtures significantly reduced the yield stress and enhanced the flowability, which was also why B2 obtained the highest flowability. The synergistic effect of the USL and LS increased the flowability of the UHPC. In order to further characterize the influence of the yield stress on the slump flow of the UHPC, the corresponding relation between them is shown in Figure 6B. On the whole, the yield stress had a negatively correlation with the slump flow. A high yield stress hindered the fluidity of the UHPC mixture.

### 3.3. Compressive Strength

Figure 7 illustrates the changes in the compressive strength of the UHPC containing the USL or LS. The compressive strength of the UHPC at 7 d obtained a similar trend of a compressive strength change at 28 d, which was increased first and then decreased with an increase in the LS. With 20% USL and 10% LS, B2 obtained the highest compressive strength of 103.7 MPa and 146.3 MPa at 7 d and 28 d, respectively. The compressive strength of B2 at 28d increased by 11.1%, 4.4%, and 11.2% compared to the Ref, USL30, and B6, respectively, which illustrated that the appropriate ratio of LS to the USL in the UHPC not only improved the flowability of the mixture but also enhanced the compressive strength of the UHPC. The large LS particle extended the skeleton space of the paste, the cement particles filled the particle space of the skeleton space of the LS, and the USL particles filled the space between the cement particles. The rational particle distribution in the ternary cement blends containing the USL and LS increased the free water and released part of the entrapped air, which enhanced the hydration of the cement and decreased the porosity, leading to an increase in the compressive strength [10,40]. In addition, the low water ratio and high alkali environment led to a rupture of the Si–O and Al–O bonds within the USL particles. These fractures promoted the dissolution of the Si and Al ions and accelerated the pozzolanic reaction with calcium hydroxide(CH), which improved the compaction and compressive strength of the UHPC [41,42]. The LS particles reacted with the aluminate phase from the USL and the cement. When the mass percentage of the USL to LS was 20:10, there was a sufficient aluminum phase to react with the LS, which formed the ettringite, Hc, and Mc [2,14,43]. They also attributed a higher strength and illustrated the synergistic action between the USL and LS-enhanced compressive strength.

### 3.4. XRD Patterns

The XRD patterns of the mixtures hydrated at 1 d, 3 d, and 28 d are illustrated in Figure 8. The CH was clearly observed, while the opposite was true for the C–S–H with a poor crystallinity. The XRD patterns found that the pastes characteristic phase (ettringite (AFt), Hc, and Mc) evolutions varied considerably with the different content of the USL and LS. At 1 d, the CH peak increased with the LS content due to the LS particles providing plenty of nucleation sites and promoting the hydration of the cement and the formation of CH crystals [40]. With the ongoing hydration, the pozzolanic reaction between the USL particles and the CH decreased the CH peak [44]. Apart from providing a nucleation site, the LS reacted with the Al phase released from the USL and the cement, which formed the AFt, Hc, and Mc hydration products. After 1 d of hydration, the USL30 obtained the lowest peak intensity of the AFt, but incorporating LS, this peak intensity increased. B2 achieved a higher peak intensity of AFt than the other ternary blends. A similar pattern was observed in the peak intensity of the Mc compared to the AFt. This result illustrated that the LS promoted the USL and cement hydration and increased the formation of the AFt. Little to no Hc and Mc were observed throughout the hydration of the Ref. The Mc was found in the binary or ternary blends during the whole hydration. The content of the Hc was lower at 1 d, higher at 7 d, and lower at 28 d in the binary or ternary blends, which demonstrated that both the formation and transformation of the Hc to Mc were closely related to the LS during the hydration [43,45]. The little Hc existence in B2 may be related to the restrictions of the carbonate transport ions by the blockage of the pore structure due to the Mc [46,47]. The Mc formation instead of the Hc due to the reaction of the LS compensated for the consumption of the CH by the Hc formation [48], which illustrated that the appropriate ratio of LS to the USL in the pastes improved the density of the concrete and also demonstrated the synergistic effect between the USL and LS.

### 3.5. Hydration Products

The SEM pictures of the Ref, B2, B6, and USL30 at 1 d and 28 d are shown in Figure 9. After 1 d of hydration, the microstructure of the Ref seemed denser than the other samples due to the dilution effect of the admixtures. Due to the nucleation, few hydration products can be seen on the surface of the USL and LS particles. After curing for 28 d, the microstructure of the paste was denser. Especially in B2, the USL and LS acted as fillers to increase the density of the paste and produced plenty of hydration products to fill the pore structure. Moreover, the USL and LS particles were covered by the hydration products. In B6 and USL30, a relatively loose space was observed, and relatively small amounts of the hydration products were found around the USL or LS particles.

The backscattered electron (BSE) microscope pictures of the Ref, B2, B6, and USL30 pastes at 28 d are compared and shown in Figure 10. The brighter parts of the picture were the non-hydrated cement particles, the grey particles were LS, the fine fragmented particles were the USL, and the black particles were the pores inside the paste. The remaining part was a mixture of the hydration products of the C–S–H with a similar grey level. As can be seen, except for some minor pores caused by the entrapped air, the UHPC paste was very dense. Incorporating the USL or LS in the UHPC reduced the cement content, decreased the number of pores, and improved the density of the paste. The synergistic effect of the appropriate proportions between the USL and LS achieved a good particle packing and attributed a higher strength.

### 3.6. Structure of the C–S–H

To explore the synergistic action between the USL and LS on the C–S–H structure in the UHPC paste, ^29^Si NMR was supplied. The quantitative results of the NMR signals were calculated and fitted, as shown in Figure 11 and Table 4. According to the experimental and simulated spectra of the UHPC paste, the Q^0^ signal corresponding to the clinker minerals in the samples containing the USL or LS exhibited a slight drop and a leftward shift compared to the Ref [49]. Meanwhile, the Q^1^ and Q^2^ represented a silicate tetrahedra, which appeared in the end and middle of the C–S–H chain, respectively [50]. Q^2^(1Al) denoted that an Si atom in the C–S–H chain was replaced by an Al atom [51], which changed the aluminum to silicon (Al/Si) ratio. In addition, the mean silicate tetrahedra chain length (MCL) was also influenced by the substitution of Al atoms. The MCL and Al/Si were obtained by calculating the relative areas of the individual after the peak fitting. They C–S–H structures were determined using Equations (5) and (6) [49,52]:(5)$$Al/Si=\frac{{\frac{1}{2}Q}^{2}\left(1Al\right)}{Q^{1}+Q^{2}\left(0Al\right)+Q^{2}\left(1Al\right)}$$
(6)$$MCl=\frac{2\left[Q^{1}+Q^{2}\left(0Al\right)+{\frac{3}{2}Q}^{2}\left(1Al\right)\right]}{Q^{1}}$$
where Q^(n)^ represents the relative areas of Q^(n)^ in the UHPC paste.

According to Table 4, compared to the Ref, the MCL of USL30, B2, B4, and B6 increased to 5.80, 6.34, 5.89, and 5.68, respectively. Meanwhile, the Al/SI of USL30, B2, B4, and B6 increased to 0.196, 0.182, 0.167, and 0.140, respectively. Incorporating the USL or LS increased the MCL, and the Al/Si may be related to the two factors. On the one hand, the amorphous silicon oxide and aluminum oxide of the USL were corroded in high alkaline environments due to the pozzolanic reaction, which led to high levels of silicon oxide and aluminum oxide being released from the glassy phases and participating in the pozzolanic reaction. Those free Si and Al ions increased the polymerization and MCL of the C–S–H and Al/Si ratio. Furthermore, the high Al content in the USL increased the probability that the Al would be inserted into the silicate chain segments and replace the Si in the C–S–H chain, which enhanced the polymerization of the C–S–H. The Q^2^(1Al) and MCL in B2 were higher and more substantial than the other pastes, which illustrated that the USL reacted with the CH and formed a large number of C–S–Hs and enhanced the polymerization of the C–S–H [53]. High levels of silicon oxide and aluminum oxide were released from the USL and LS and reacted with the Al phases, forming the Mc, which consumed the Al ions and decreased the Al ions concentration, promoting the further USL hydration. This demonstrated that the synergistic effect of the USL and LS increased the MCL and polymerization degree of the C–S–H. The quantity of literature has shown that longer MCLs were often accompanied by higher compressive strengths [52,54]. The hypothesis supported that the longest MCL in B2 may result in enhancing the higher compressive strengths of the UHPC.

### 3.7. Pore Structure

The effect of the USL or LS on the pore size distribution and total porosity of the UHPC is shown in Figure 12. According to Figure 12, incorporating the USL or LS in the UHPC decreased the total porosity of the paste and led to the peak of the size distribution dropping and shifting left. More details of the pore characteristic parameters are illustrated in Table 5. The total porosity of USL30, B1, B2, and B6 were 7.25%, 6.47%, 7.33%, and 7.69%, which decreased by 7.4%, 17.4%, 6.4%, and 1.8%, respectively, compared to the Ref. B2 obtained the lowest porosity and the critical diameter of the capillary (CDC) in all the UHPC pastes. This was mainly due to the physical and chemical synergy of the USL and LS. For the physical synergy of the USL and LS, the appropriate proportions filled the large capillary and released the entrapped air, decreasing the CDC and porosity. For the chemical synergy of the USL and LS, the pozzolanic reaction of the USL with the CH formed an additional C–S–H and the reaction of the LS compensated for the consumption of the CH by the Mc formation, which contributed to the increase in the gel pores and refined the pore structures [55]. The physical and chemical synergy of the USL and LS produced a dense microstructure and improved the pore size distribution of the UHPC.

### 3.8. Pore Surface Fractal Dimension (Ds)

The synergy effect of the USL and LS decreased the porosity of the UHPC and refined the pore structures. Their synergy effect on the complexity and heterogeneity of the pores in the UHPC system cannot be adequately evaluated by the porosity. The fractal dimension was selected to better understand the synergy effect on the pore structures. The Ds values of the UHPC containing the USL or LS at 28 d are shown in Table 6. All the R^2^ of the UHPC were close to 1, which suggested that the results were credible and accurate. In addition, a Ds value lower than 2 or higher than 3 was meaningless, according to the fractal theories [56,57]. The Ds values of all the UHPC varied between 2.589 and 2.683, which satisfied the calculations of the fractal theory [18,58]. The Ds value of B2 obtained the most considerable value of 2.683 due to the synergy effect of the USL and LS. Plenty of literature demonstrated that the percentage of 3–20 nm pores in the concrete was crucial for determining the resistance to the chloride ion penetration and shrinkage as well as the compressive strength of the concrete [58,59]. The volume fraction with pores less than 20 nm was named *V*_3–20 nm_, containing the fractions of the medium capillary and gel pore. Additionally, this relationship between the Ds and the porosity, CDC, *V*_3–20 nm_, and compressive strength of the UHPC are shown in Figure 13. According to Figure 13A B, the relationship between the Ds and the porosity and CDC illustrated a negative correlation with high R^2^ values of 0.988 and 0.966, respectively. For Figure (C) and (D), a similar positive correlation was obtained between the Ds and the *V*_3–20 nm_ and compressive strength of the UHPC with high R^2^ values of 0.991 and 0.992, respectively. They demonstrated that the pore structure’s complexity and irregularity could be characterized and represented by the Ds. Furthermore, the Ds maintained a positive correlation with the compressive strength, and more importantly, the synergistic effect resulted in a higher compressive strength of the UHPC, which demonstrated that the Ds parameter remained strongly correlated with the synergistic effect. The pozzolanic reaction and filling effect, as well as the synergistic effect between the USL and LS, refined the pore structures and increased the *V*_3–50 nm_ of the UHPC, which enhanced the changes in the pore structure complexity and irregularity. Compared to the relationship between the Ds and the pore structure parameters, as well as the compressive strength, the fractal theory can be conducted to understand the synergistic effect between the USL and LS on the properties of the UHPC. Thus, a new metric called the Ds value may be used to assess the synergistic effect of the UHPC.

## 4. Conclusions

The key observation in this study was the confirmation of the synergetic interaction between LS and the USL on the properties of the UHPC. From the results presented in this paper, the following conclusions can be drawn.

The synergistic effect between the USL and LS in the UHPC resulted in a decrease in the yield stress and fluid flow index, and an increase in the workability and compressive strength. Among the different USL to LS mass ratios tested, B2 with a ratio of 2:1 obtained the highest compressive strength, maximum slump flowability, and lowest yield stress.The synergistic interaction between the USL and LS improved the homogeneity of the paste, promoted cement hydration, facilitated the formation of the ettringite and monosulfate phases, and increased the chain length of the calcium-silicate-hydrate (C–S–H) gel. These effects refined the pore structure and led to a denser microstructure in the UHPC.The fractal dimension (Ds) of the ultra-high performance concrete (UHPC) was found to be strongly correlated with its pore structure and compressive strength, suggesting that the fractal theory can be used to better understand the synergistic effect between the USL and LS on the properties of the UHPC. Therefore, the Ds value may be used as an indicator to assess the synergistic effect of the USL and LS in the UHPC.

## Figures and Tables

**Figure 1 materials-16-02281-f001:**
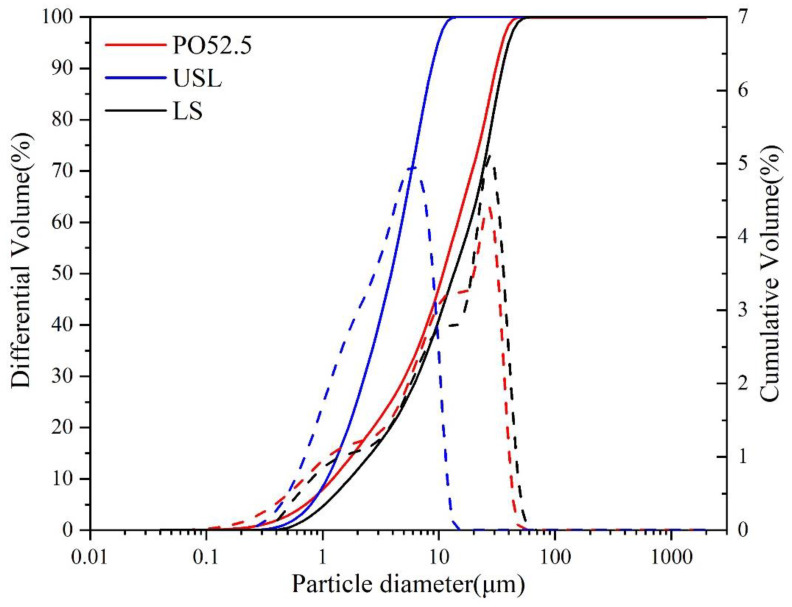
Particle distribution of P·O 52.5, USL, and LS.

**Figure 2 materials-16-02281-f002:**
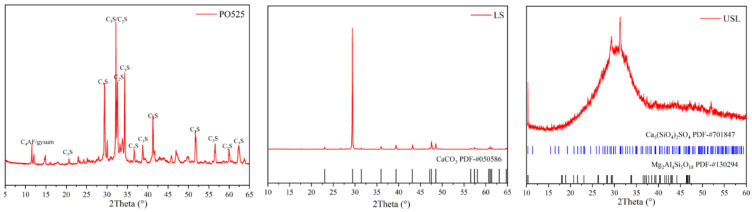
XRD analysis of P.O 52.5, HPS, and LS.

**Figure 3 materials-16-02281-f003:**
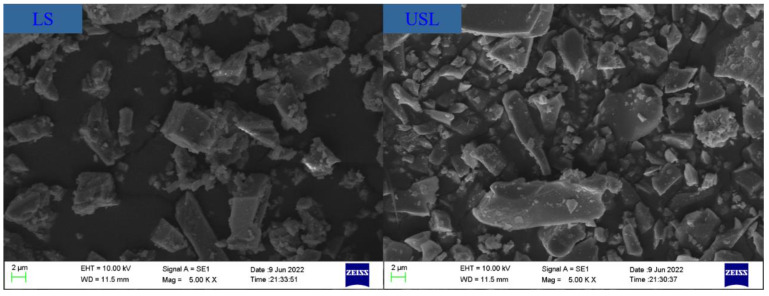
SEM pictures of USL and LS.

**Figure 4 materials-16-02281-f004:**
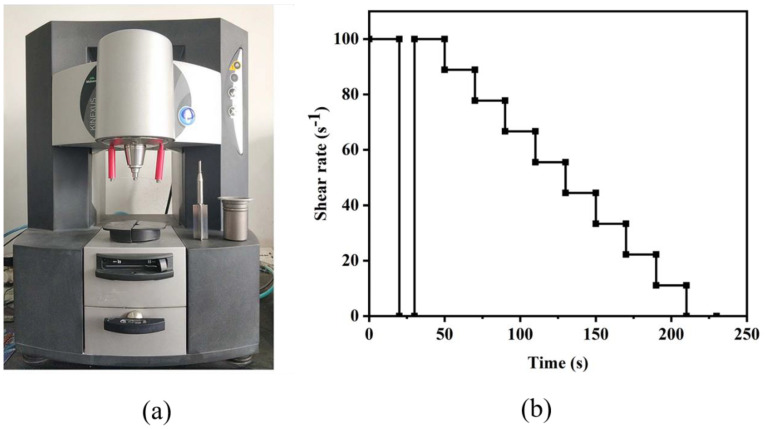
Malvern rheometer (**a**) and rheological test protocol (**b**).

**Figure 5 materials-16-02281-f005:**
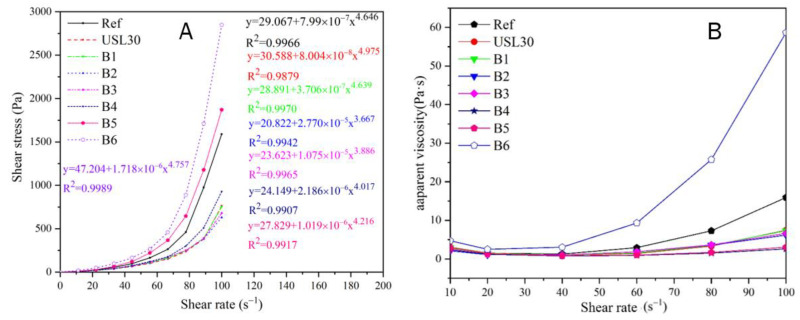
Rheology curves of the cement slurry with different USL or LS content. (**A**) shear stress (**B**) apparent viscosity.

**Figure 6 materials-16-02281-f006:**
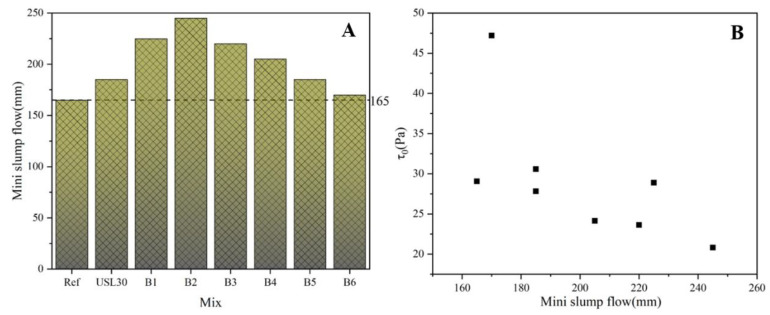
Mini slump flow of the UHPC (**A**) and the relationship between the mini slump flow and yield stress (**B**).

**Figure 7 materials-16-02281-f007:**
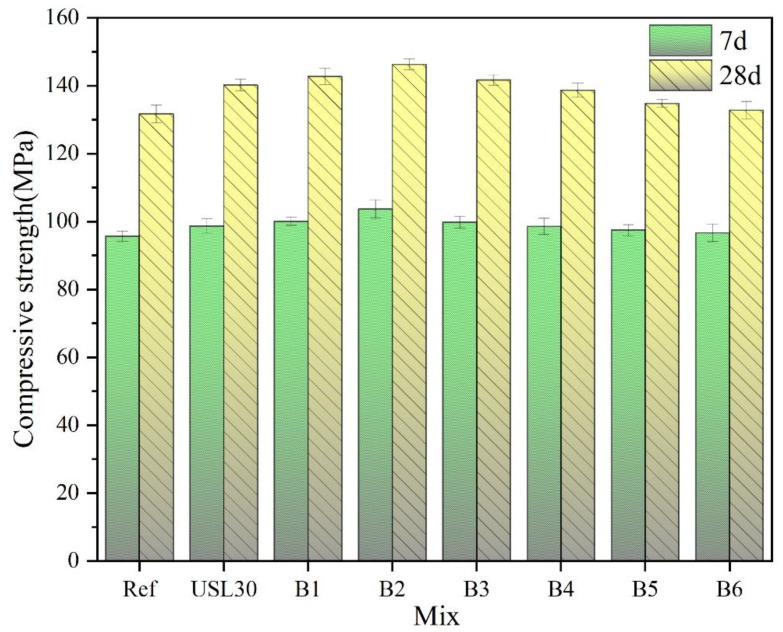
Changes in the compressive strength of the UHPC containing USL or LS.

**Figure 8 materials-16-02281-f008:**
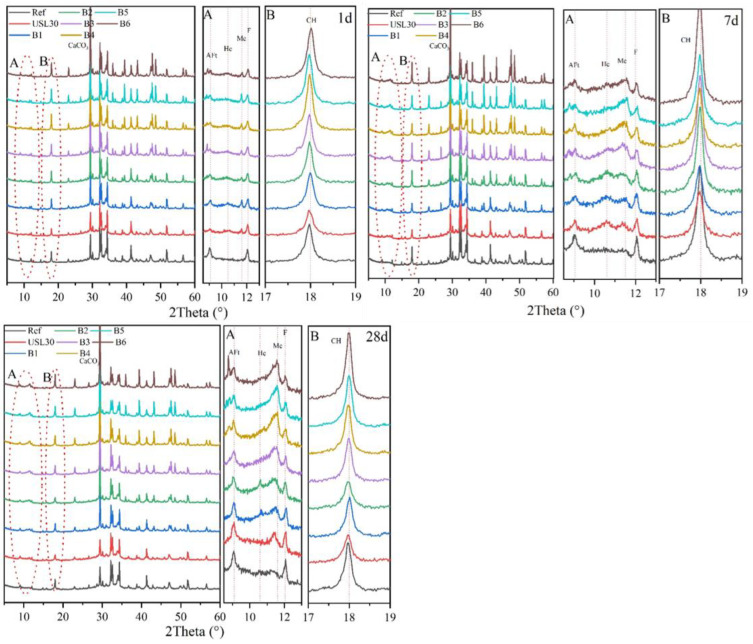
XRD patterns of the blends at 1 d, 3 d, and 28 d.

**Figure 9 materials-16-02281-f009:**
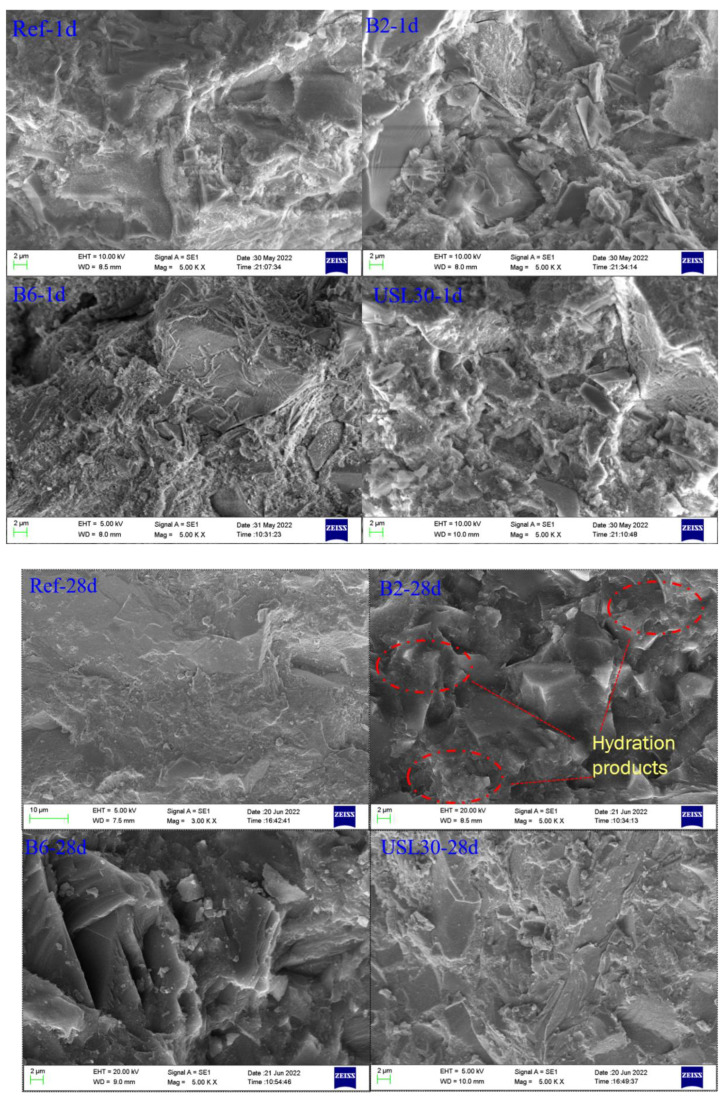
SEM pictures of the Ref, B2, B6, and USL30 pastes at 1 d and 28 d.

**Figure 10 materials-16-02281-f010:**
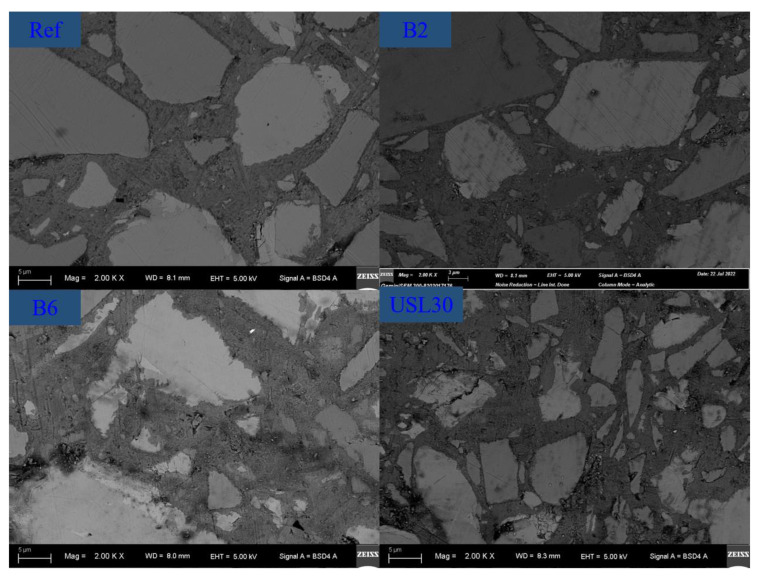
BSE images of the Ref, B2, B6, and USL30 pastes at 28 d.

**Figure 11 materials-16-02281-f011:**
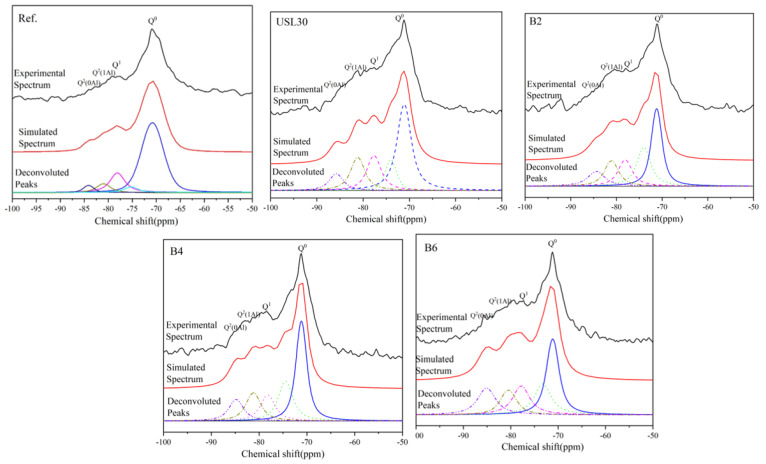
^29^Si NMR spectra of the Ref, USL30, B2, B4, and B6.

**Figure 12 materials-16-02281-f012:**
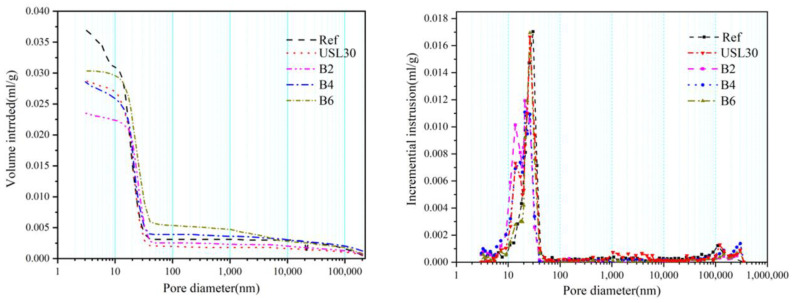
MIP results with USL or LS at 28 d.

**Figure 13 materials-16-02281-f013:**
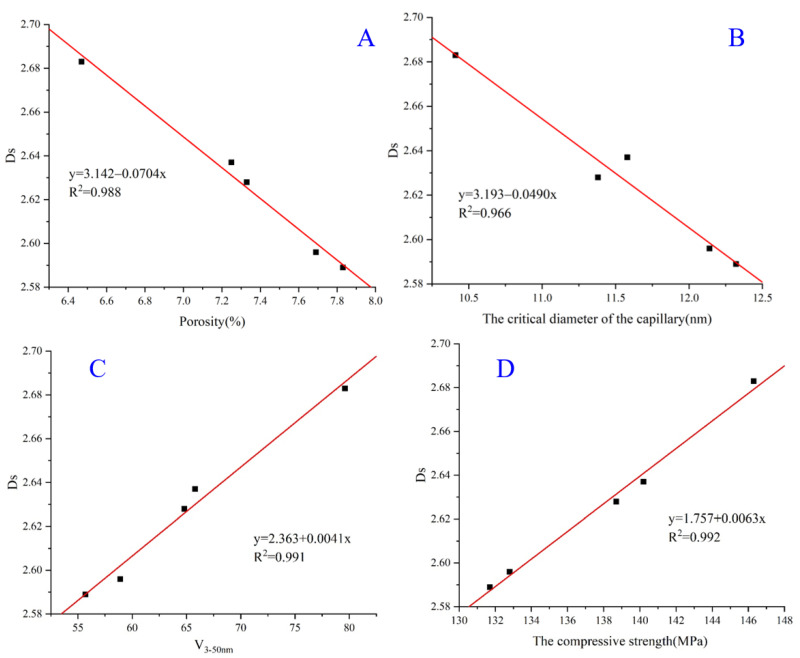
Relationship between the Ds and the (**A**) porosity, (**B**) CDC, (**C**) *V*_3–20 nm_, and (**D**) compressive strength of the UHPC.

**Table 1 materials-16-02281-t001:** Chemical composition of the binders (wt.%).

Oxide Formula	Na_2_O	MgO	Al_2_O_3_	SiO_2_	P_2_O_5_	SO_3_	Cl^−^	K_2_O	CaO	TiO_2_	MnO	Fe_2_O_3_	LOI
USL	0.66	9.37	16.82	27.39	0	2.60	0.07	0.32	36.31	0.91	0.37	0.33	1.20
LS	0.16	1.11	1.48	3.19	0.06	0.08	0.02	0.29	59.42	0	0.07	1.22	40.58
P·O52.5	0.50	4.94	8.06	22.89	0.10	3.55	0.11	0.73	52.84	0.38	0.13	2.39	3.27

**Table 2 materials-16-02281-t002:** Mixture proportioning the UHPC mixtures (kg/m^3^).

	P·O 52.5	LS	USL	Water	SP	Sand	Steel Fiber
Ref	1150	0	0	207	17.25	1150	152
USL30	805	0	345	207	17.25	1150	152
B1	805	57.5	287.5	207	17.25	1150	152
B2	805	115	230	207	17.25	1150	152
B3	805	172.5	172.5	207	17.25	1150	152
B4	805	230	115	207	17.25	1150	152
B5	805	287.5	57.5	207	17.25	1150	152
B6	805	345	0	207	17.25	1150	152

**Table 3 materials-16-02281-t003:** Herschel–Bulkley parameters and of the plastic viscosity paste with different LS and USL dosages.

	**Herschel**–**Bulkley Parameters**			R^2^	Plastic Viscosity (Pa·s)
	τ_0_/Pa	*k*	*n*		
Ref	29.067	7.99 × 10^−7^	4.646	0.9966	7.065
USL30	30.588	8.00 × 10^−7^	4.975	0.9879	3.067
B1	28.891	3.71 × 10^−7^	4.639	0.9970	3.180
B2	20.822	2.77 × 10^−5^	3.667	0.9942	0.0316
B3	23.623	1.08 × 10^−5^	3.886	0.9965	0.0326
B4	24.149	2.19 × 10^−6^	4.017	0.9907	0.118
B5	27.829	1.019 × 10^−6^	4.216	0.9917	0.133
B6	47.204	1.7818 × 10^−6^	4.757	0.9989	2.584

**Table 4 materials-16-02281-t004:** Deconvolution results for the ^29^Si NMR spectra (%).

NO.		Q^0^	Q^1^	Q^2^(1Al)	Q^2^(0Al)	Al/Si	MCL
Ref	Peak (ppm)	−70.8	−78.1	−81.0	−84.1	0.107	3.25
	Area (%)	67.18	21.53	6.79	3.31		
USL30	Peak (ppm)	−71.1	−77.7	−81.2	−85.8	0.196	5.80
	Area (%)	55.93	18.16	17.22	8.69		
B2	Peak (ppm)	−71.2	−78.1	−81.1	−84.4	0.182	6.34
	Area (%)	57.64	15.79	15.41	11.15		
B4	Peak (ppm)	−71.15	−78.2	−81.2	−84.2	0.167	5.89
	Area (%)	57.58	16.82	14.16	11.44		
B6	Peak (ppm)	−71.15	−77.8	−80.5	−85.1	0.140	5.68
	Area (%)	55.85	17.73	12.40	14.03		

**Table 5 materials-16-02281-t005:** Pore distribution of the UHPC.

Samples	Average Pore Diameter (nm)	Median Pore Diameter (nm)		Critical Diameter of the Capillary (nm)	Porosity (by Volume) (%)	Pore Distribution (%)		
		**By Area Method**	**By Volume Method**			<10 nm	10–100 nm	>100 nm
Ref	20.16	19.63	17.76	12.32	7.83	5.52	63.68	29.8
USL30	17.87	17.77	20.80	11.58	7.25	6.51	86.2	7.29
B2	18.1	17.25	21.63	10.41	6.47	8.41	78.21	13.38
B4	22.63	21.62	25.49	11.38	7.33	5.64	83.49	10.87
B6	23.80	20.59	25.65	12.14	7.69	2.56	79.82	17.62

**Table 6 materials-16-02281-t006:** Ds of the UHPC paste at 28 d.

No.	Ds	R^2^
Ref	2.589	0.996
USL30	2.637	0.994
B2	2.683	0.997
B4	2.628	0.992
B6	2.596	0.989

## Data Availability

Not applicable.

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
