# Peer review of "The Synergistic Effects of Ultrafine Slag Powder and Limestone on the Rheology Behavior, Microstructure, and Fractal Features of Ultra-High Performance Concrete (UHPC)"

_materials, 2023, doi:10.3390/ma16062281_

Round 1

Reviewer 1 Report

The mixture compositions (Table 2) is not correct because the densities of the raw materials were not well considered.

Author Response

Thank you very much for your approval and comments on our work. The paper has been carefully revised according to your valuable suggestions. Relevant amendments have been added.

Reviewer 2 Report

1) The introduction is well-written and contextualizes the study's relevance.

2) In Fig 2, identify the peaks and insert the database and the code of crystalline phases.

3) In section 2.2., enter details about the mixing procedure, mixing speed, time, etc.

4) What geometry was used in the rheometry test? What is the gap used? Test temperature? How did the rate of acquisition of shear stress occur? How long was the shear rate maintained for the determination of shear stress?

5) Insert the shear rate x viscosity curves, which can better show the shear-thickening response.

6) “Consequently, the plastic viscosity could not be given due to the nonlinear relationship between the shear stresses and the shear rates.” Plastic viscosity was not calculated. To check. In fact, the Herschel-Bulkley fit does not provide this parameter. One suggestion may be to use the De Larrard Equation.

7) The discussion of rheometric results needs to be revised entirely. Authors need to justify behaviours (e.g., shear stress behaviour with increasing LS and USL contents). Furthermore, it is necessary to compare the results obtained with those described in the literature.

8) What is the correlation between yield stress (determined in the rheometry test) and mini-slump??

9) “Rational particle distribution in ternary cement blends containing USL and LS increased the free water and released part of the entrapped air, which enhanced the hydration of cement and decreased the porosity, leading to the increase in compressive strength” Explain this sentence better.

10) Fig 8 is too small, making it impossible to visualize the data. Increase the size of the graphics.

11) “The content of Hc was lower at 1d, higher in 7d, and lower at 28d, respectively, in binary or ternary blends, which demonstrated both the formation and transformation of Hc to Mc were closely related to LS during hydration. The little Hc existence in B2 may be related to the restrictions of the carbonate ions transport ions by the blockage of the pore structure due to the Mc. The Mc formation instead of Hc due to the reaction of LS compensated for the consumption of CH by the Hc formation, which illustrated the appropriate ratio of LS to USL in the pastes improved the density of the concrete and also demonstrated the synergistic effect between the USL and LS.” Insert references to support this information.

12) “Moreover, the USL and LS particles were covered by the hydration products…” Indicate in the figure to facilitate visualization.

13) Improve work conclusions. These are very generic sentences. Also, insert a close on the overall contribution of the work.

Author Response

(The authors gave the same response as above.)

Reviewer 3 Report

I have attached comments for the authors.

Author Response

(The authors gave the same response as above.)

Round 2

Reviewer 1 Report

The corrections asked by me were not made.

Author Response

Thank you very much for your approval and comments on our work. The paper has been carefully revised according to your valuable suggestions. 

Reviewer 2 Report

Accept

Author Response

Thank you again. Your insightful comments and suggestions played a crucial role in improving the quality of the manuscript, and I appreciate the time and effort you put into the review process. Your feedback helped me to refine my ideas and to communicate my research findings more clearly and effectively.